# The Effects of Feeding Milled Rapeseed Seeds with Different Forage:Concentrate Ratios in Jersey Dairy Cows on Milk Production, Milk Fatty Acid Composition, and Milk Antioxidant Capacity

**DOI:** 10.3390/life13010046

**Published:** 2022-12-23

**Authors:** Daniel Mierlita, Anita Santa, Stefania Mierlita, Stelian Vasile Daraban, Mihai Suteu, Ioan Mircea Pop, Olimpia Smaranda Mintas, Adrian Maximilian Macri

**Affiliations:** 1Department of Animal Science, Faculty of Environmental Protection, University of Oradea, 1 University St., 410087 Oradea, Romania; 2Doctoral School of Agricultural Engineering Sciences, University of Agricultural Sciences and Veterinary Medicine, 3-5 Manastur St., 400372 Cluj-Napoca, Romania; 3Department of Accounting and Audit, Faculty of Economics and Business Administration, Babeş-Bolyai University, 58-60 Teodor Mihali St., 400372 Cluj-Napoca, Romania; 4Department of Technological Science, Faculty of Animal Science and Biotechnologies, University of Agricultural Sciences and Veterinary Medicine, 3-5 Manastur St., 400372 Cluj-Napoca, Romania; 5Department of Animal Nutrition, Faculty of Food and Animal Sciences, Ion Ionescu de la Brad University of Life Sciences Iasi, 3 Mihail Sadoveanu Alley, 700490 Iasi, Romania; 6Department of Animal Nutrition, Faculty of Veterinary Medicine, University of Agricultural Sciences and Veterinary Medicine, 3-5 Manastur St., 400372 Cluj-Napoca, Romania

**Keywords:** milk, FA, health lipid indices, rapeseed, concentrate level, total antioxidant capacity

## Abstract

We aimed to evaluate the effects of milled rapeseed (MR) supplementation of low- or high-concentrate diets on milk production and composition, fatty acids (FAs) profile, and antioxidant capacity. Sixteen Jersey dairy cows were used in a 4 × 4 Latin square design, for four periods of 4 weeks, and assigned to four treatments as a 2 × 2 factorial design. Dietary treatments consisted of iso-nitrogenated total mixed rations with high (65:35; LC—low concentrate) or low (50:50; HC—high concentrate) forage:concentrate (FC) ratios, supplemented with MR to provide 30 g oil/kg dry matter (DM) (LR and HR), or without MR supplement (L and H). Increasing the proportion of concentrates led to an increase in DM intake (DMI), net energy (NE_L_) intake, and milk production, but milk fat and protein content decreased. Supplementing diets with MR led to an increase in NE_L_ intake and milk production, but did not affect DMI and milk composition. Diets supplemented with MR caused a decrease in the concentration of FAs with atherogenic effect and the increase in the level of FAs beneficial for human health (C18:1 *cis*-9, C18:1 *trans*-11, and C18:3 n-3), while the decrease in the FC ratio had a negative effect on omega-3 FAs. An improvement in the antioxidant capacity of milk was observed with diets with the high FC ratio but also by supplementing the feed with MR. These results could contribute to the development of effective strategies to improve the nutritional quality of milk without affecting the productive performance of cows.

## 1. Introduction

Nowadays, many consumers are aware of the relation between diet and health. A major interest in modifying milk fatty acids (FAs) composition, aiming to enhance the health status of consumers, was noticed in the last two decades [1,2,3]. Such changes are intended to decrease the proportion of saturated FAs (SFAs) in milk fat composition, and to increase the proportion of healthy FAs, such omega-3 FAs (n-3 FAs), conjugated linoleic acid (CLA, *cis*-9, *trans*-11 C18:2), oleic acid (OA, C18:1 *cis*-9), and vaccenic acid (VA, C18:1 *trans*-11) [4]. These FAs play an important role in human health by reducing the risk of chronic diseases [5], combatting cancer, preventing atherosclerosis, reducing body fat, and modulating inflammatory and/or immune responses [6].

The FAs profile of milk depends on several factors; nutritional management being one of most important [7,8]. Supplementing cow diets with different sources and types of fat is an effective way of manipulating the FAs profile of milk fat [3,9,10]. Previous research demonstrated that supplementing cow diets with vegetable oils led to an increase in n-3 FAs and CLA concentration in milk fat [7,11]. Oils rich in linoleic acid (LA, C18:2 n-6; e.g., sunflower oil) were more effective in increasing CLA in milk fat than oils rich in α-linolenic acid (ALA, C18:3 n-3; e.g., linseed oil), which was more effective in increasing n-3 FAs in milk fat [3,12]. Only a limited number of studies have examined how oils which are OA-rich (oleic acid, C18:1 *cis*-9) (e.g., rapeseed oil) affect lactation performance, the FAs content of milk [13,14,15], and enteric methane emissions in dairy cows [16,17]. The form of rapeseed inclusion in dairy cow diets influences the lactation performance and the FAs profile of milk fat. The research carried out by Givens et al. [15] demonstrated that milk production was higher when rapeseed was ground, compared to the use of whole rapeseeds. The bioavailability of lipids was found to be higher when rapeseed was ground and resulted in a milk FAs profile similar to that determined by rapeseed oil, reducing milk SFAs and increasing *cis*-MUFA concentrations, and a lower increase in *trans*-FAs [15]. However, direct comparisons of the efficacy of diet supplementation with fats rich in OA (rapeseed oil or rapeseed seeds), starting from different FC ratios, on milk production, milk fat FAs profile, and milk TAC, are scarce.

In dairy cow rations, modifying the forage to concentrate (FC) ratio or supplementation with unsaturated fats affects rumen fermentation pathways, resulting in different volatile fatty acids profiles and, thereby, different milk FAs composition [18]. Increasing the proportion of concentrates in cow diets containing vegetable oils often causes a decrease in milk fat content [17,19,20] and an increase in *trans* FAs concentrations, that have an inhibitory effect on the de novo synthesis of some FAs [18,21,22].

Increasing the concentration of nutritionally beneficial FAs in milk also increases the risk of milk fat oxidation, leading to changes in the nutritional and dietary properties of milk [1]. To maintain high milk quality, the concentration of antioxidants should therefore be increased [23], for them to ensure an increased antioxidant capacity of milk [2,24]. Higher concentrations of milk antioxidants (α-tocopherol, β-carotene, and retinol) have been reported in cows fed diets based on grass silage compared to diets rich in concentrates [25], but also by supplementing the diet with oilseeds [26].

Only a limited number of studies, with heterogeneous results, assessed the optimal FC ratio in diets supplemented with vegetable fats with the aim of obtaining milk with a healthier FAs profile (for human consumers), without affecting the productive performances of cows [18,19]. Consequently, the aim of this study was to evaluate the effects of two FC ratios and supplementation with ground rapeseed on milk production and composition, FAs profile, and the TAC of milk obtained from Jersey cows.

## 2. Materials and Methods

All experimental procedures were approved by the Ethics Committee of the Faculty of Environmental Protection (5 of 25/07/2022) within the University of Oradea (Romania), in accordance with the European Union Directive 2010/63/EU on the use of animals for scientific purposes.

### 2.1. Animals, Diets, and Experimental Design

Sixteen multiparous Jersey cows (mean at the start of the experiment ± standard deviation: cow weight: 491.0 ± 74.2 kg, days in lactation: 116 ± 24.6 d, mean milk yield: 23.2 ± 1.9 kg milk/d) were used in a replicated (n = 4) 4 × 4 Latin square, with a 2 × 2 factorial arrangement of treatments. Each experimental period consisted of 28 d, of which the first 21 d were for adaptation to the new diet, while in the last 7 d the productive performances were recorded and samples were taken for laboratory analyzes (Figure 1).

All diets were formulated to meet the nutritional requirements of dairy cows [27], for a milk yield of 26.0 kg/d and a consumption of 21.8 kg DM/head/d, considering a refusal of 10%.

The grass silage was prepared from perennial ryegrass and white clover plants, which after mowing were left to wither for 24 h, then harvested, chopped, and ensiled naturally, without silage additives.

The concentrates were formulated as supplements so that the experimental diets were isonitrogenous and isoenergetic. The inherent difference between the diets in protein content was minimized by adjusting the level of dietary protein with rapeseed meal. No adjustment was made to the L and H diets to compensate for the fat added by the MR.

The experimental diets were balanced for crude protein (CP) content. Starch, fat, neutral detergent fibers (NDF), and acid detergent fibers (ADF) contents were allowed to differ between diets in light of the different to FC ratios and MR supplementation.

Unlike other oleaginous seeds, rapeseed seeds are very resistant to degradation both in the rumen and in the small intestine of cows [12], and therefore their processing is necessary. Thus, the rape seeds were ground separately with a hammer mill equipped with a 2.8 kw motor and a 661 × 67 mm sieve with 2 mm hole diameters. Ground rapeseeds were introduced into the concentrate mixture.

Cows were housed in stalls, separated by study groups, on a commercial farm. The food was prepared daily and administered twice a day, after the morning and afternoon milking, in amounts that allowed a refusal of up to 10%.

### 2.2. Measurements, Sample Collection, and Chemical Analyses

TMR intake and milk production were recorded throughout each experimental period, but only measurements taken in the last week (between days 22 and 28) were used for the statistical analysis.

Cows were milked twice a day at 07:00 and 18:00 h, and individual milk production was recorded using a volumetric milk meter. Milk samples were taken for 3 consecutive days (25, 26, and 27 d) during each experimental period. Three milk samples of 50 mL each were collected. A sample was treated with a preservative (Bromopol; D&F Control Systems, San Ramon, CA, USA) and stored at 40 °C until it was analyzed by infrared spectrophotometry [28] for fat, protein, lactose, and urea using a Milko-Scan FT+ analyzer (Foss Electric, Hillerød, Denmark). The other two samples of untreated milk were stored at −20 °C and later used for the analysis of FAs, the content of antioxidants, and the TAC, respectively.

On the same days of all experimental periods (25, 26 and 27 d), representative samples of silage, alfalfa hay, mixture of concentrates and TMR were taken daily from which composite samples were then made and stored at –20 °C until they were analyzed for DM and chemical composition. In grass silage the DM content was corrected for the loss of volatile substances [29]. Forage, concentrate, and TMR samples were dried at 60 °C for 24 h to determine the DM content [29]. After drying, the samples were ground with a Cyclotec mill (Tecator 1093, Hoganas, Sweden), using a 1.0 mm screen. The nitrogen content was estimated by the Kjeldahl technique, using an Vapodest Distillation Systems (VAP 400; C. Gerhardt GmbH & Co. KG, Königswinter, Germany) (Method 5983-2). The CP content was calculated by multiplying the N content by 6.25 [30]. Crude fat (CF) was determined by extraction with petroleum ether (boiling point, 40 to 60 °C) using an Soxhlet laboratory extractor (EV6 AII/16; C. Gerhardt GmbH & Co. KG, Germany) (AOCS Method) [31]. The ash content was determined by heating 2 g of the dried sample in a silica dish at 580 °C for 8 h, using an calcination furnace (Nabertherm LVT 3/11, Lilienthal, Germany) (Method ISO 5984) [32]. NDF and ADF were determined according to the method of Van Soest et al. [33], using an Ankom 220 Fibber Analyser (ANKOM Technology Corp., Fairport, New York, NY, USA). For the measurement of feed starch, the combined action α-amylase and amyloglucosidase to hydrolyze starch to glucose was used, followed by glucose determination with a glucose oxidase/peroxidase reagent (AOAC Method No 996.11) [34]. Energy content was estimated according to NRC [27].
NE_L_ (Mcal/kg DM) = 0.0245 × TDN (%) − 0.12.(1)
where TDN—Total Digestible Nutrients:TDN (% of DM) = (digestible OM − digestible CF) + (digestible CF × 2.25).(2)
where digestible OM (organic matter) and CF (crude fat) are expressed as a percentage of DM. To establish the apparent digestibility of nutrients was using the NRC tables [27].

FA analysis of milk samples was performed as reported by Santa et al. [2], in two stages: transesterification/methylation of FAs and quantification of FA methyl esters (FAME) by gas chromatography (GC). The extraction of fat from milk was done according to the method described by Folch et al. [35] and the preparation of methyl esters of FAs was conducted by transmethylation of methyl alcohol. The lipid extract was solubilized in a volume of 3 mL of chloroform, next 3 mL of benzene, 3 mL of BF3, and 1.5 mL of methanol being added. The mixture was heated to 80 °C for 2 h, afterwards 6 mL hexane and 3 mL of distilled water were added. After that, the sample was homogenized using a vortex. The supernatant was transferred to a test tube containing anhydrous Na_2_SO_4_. The samples were treated again with hexane (1 mL) and distilled water (1 mL) and mixed well by vortexing. The obtained extract was dried on a rotary evaporator, and the obtained residue was treated with hexane (1.5 mL).

FAMEs quantification by GC was carried out using a Varian 3800 GC-4000 equipment (Varian, Inc., Palo Alto, CA 94598-1675/USA), using helium as carrier gas at a constant flow rate of 1 mL/min, a CP-Sil 88 capillary column (100 m × 0.25 mm, 0.20 μm i.d.; Varian, Inc. CA 94598-1675/USA) and a flame ionization detector (260 °C). The injector temperature was at 250 °C. The initial column temperature was at 140 °C for 5 min; temperature gradient of 3 °C/min and plateau for 5 min at 200 °C. Next, a temperature gradient of 3 °C/min and a plateau for 40 min at 240 °C were used, resulting in a total running time of 83 min. FAMEs were quantified using external standards (Matreya Inc., PA.; Sigma-Aldrich Inc.), comparing their retention times with those of the pure methyl ester standard, and concentrations were calculated in g/100 g total FAs identified.

FA methyl esters from forages, concentrates, and experimental diets were obtained by methylation in a single extraction step with chloroform and methanolic sulfuric acid (2% vol/vol; [11]) and subsequently quantified by GC using the same working technique as described for the determination of the FAs profile in milk.

The extraction of retinol, α-tocopherol, and carotenes from milk was done by the standard method [34] and their quantification in milk was done by high performance liquid chromatography (HPLC). The milk sample (500 µL) was mixed with pyrogallol (400 µL) for 15 s by vortexing. Next, distilled water (1 mL) and ethanol (600 µL) were added and mixed for 1 min. The extraction was repeated twice. The samples were further treated with hexane (4 mL), mixed and centrifuged (2000× *g*) for 10 min. The organic phase (3 mL) was dried in a nitrogen atmosphere. Quantification of antioxidants was done with an HPLC (Agilent Technology series 1100, Santa Clara, CA, USA), using methanol as the mobile phase at a flow rate of 1.3 mL/min for 8 min and a Phenomenex SphereClone column (5 µm, 150 × 4.6 mm). The column was washed with MTBE:MeOH (50:50) after each determination. Chromatograms were monitored at 450 nm for β-carotene, 340 nm for retinol and 290 nm for α-tocopherol [2]. The quantification of the bioactive compounds was carried out using the regression equations of the calibration curves of the standards with a degree of purity >90% (β-carotene: C4582; *trans*-retinol: R7632, and α-tocopherol: T3251) (Sigma Aldrich, Madrid, Spain). All analyzes were performed in duplicate.

The determination of tocopherols and carotenes in food (diets and diet ingredients) was done according to the method described by [2]. A 5 g sample was treated with an extraction solvent consisting of a mixture of hexane and ethanol (1:1 *v*/*v*). The sample was filtered and then saponified with a potassium methanol solution in a nitrogen atmosphere. The two phases were separated in a separatory funnel using distilled water and diethyl ether. The organic phase after washing and drying in a stream of nitrogen gas was treated with ethyl acetate (100 µL). The quantification of tocopherols and carotenes in food was done by HPLC; the method was similar to that used for milk.

Total antioxidant activity of the milk samples was determined using the ABTS (2,20-azino-bis[3-ethylbenzothiazoline-6-sulfonic acid]) method, described by [36]. This is a spectrophotometric method for determining the ability of antioxidants to neutralize the radical cations generated from ABTS under the influence of sodium persulfate, and which determines the decrease in absorbance of the solution. Antioxidants lead to the reduction of the radical ABTS^+^ cation, causing the solution to turn blue. ABTS radical stock solution was prepared by dissolving 2,2′-azinobis(3-ethylbenzthiazoline-6-acid) in sodium persulfate (2.45 mM) and the mixture was kept in the dark for 12 h to generate the blue cation radical ABTS^+^. The ABTS solution obtained was diluted with methanol to obtain an absorbance of 0.70 ± 0.05 at 734 nm. Milk samples (0.01 mL) were treated with ABTS^+^ solution (1 mL) for 10 min at room temperature. After centrifugation (5 min, 8000 rpm) the absorbance was measured at 734 nm against the reference sample (methanol). The TAC of the milk samples was expressed in Trolox equivalent (µM Trolox/mL milk) based on the calibration curve and percent recovery.

### 2.3. Calculations and Statistical Analyses

The yield of energy-corrected milk (ECM) was calculated by the following formula [27]:ECM (kg/d) = (0.3246 × kg milk) + (12.86 × kg fat) + (7.04 × kg protein)(3)

The following formula was used for the calculation of 4% FCM of each cow [27]:FCM (kg/d) = [(0.4 × kg milk) + (0.15 × kg milk × % fat)](4)

The evaluation of the quality of milk fats, in terms of the impact on human health, was done by calculating the ratios n-6/n-3 FA, hypocholesterolemic/hypercholesterolemic FA (h/H FA), the atherogenic index (AI), the thrombogenic index (TI), and the health promotion index (HPI) [37]. The calculation of health lipid indices were used by the equation of Ulbricht and Southgate [38]:n-6/n-3 FA = (C18:2 n-6 + C18:3 n-6 + C20:4 n-6)/(C18:3 n-3 + C20:5 n-3 + C22:3 n-3 + C22:5 n-3 + C22:6 n-3);(5)
AI = (C12:0 + (C14:0 × 4) + C16:0)/MUFA + PUFA);(6)
TI = (C12:0 + C16:0 + C18:0)/[(0.5 × MUFA) + (0.5 × n-6 FA) + (3 × n-3 FA) + (n-3 FA/n-6 FA)];(7)
HPI = (n-3 PUFA + n-6 PUFA + MUFA)/[C12:0 + (4 × C14:0) + C16:0];(8)
h/H FA = (C18:1 + PUFA)/(C12:0 + C14:0 + C16:0).(9)

The evaluation of the FAs enzymatic desaturation processes that take place in the mammary gland was done by indirectly measuring the activity of the Δ^9^-desaturase enzyme, based on the ratio: (product of Δ^9^-desaturase)/(product of Δ^9^-desaturase) + (substrate of Δ^9^-desaturase). Desaturase index (DI) was calculated according to Kay et al. [39]:DI = (C14:1 + C16:1 + C18:1 *c*9+ C18:2 *c*9, *t*11)/(C14:0 + C14:1 + C16:0 + C16:1 + C18:0 + C18:1 *c*9 + C18:1 *t*11 + C18:2 *c*9, *t*11).(10)

The results obtained were analyzed by ANOVA for a 4 × 4 Latin square, with a 2 × 2 factorial arrangement of treatments, using the MIXED procedure of SAS [40], with the following model:Y_ijk_ = μ + F_l_ + E_i_ + F_l_ × E_i_ + P_j_ + C_k_ + e_ijk_,(11)
where Y_ijk_ = observation; μ = population mean; F_l_ = forage to concentrate ratio effect (l = 1, 2); E_i_ = MR effect (i = 1, 2); P_j_ = period (j = 1, 2, 3, or 4); C_k_ = random effect of cow (k = 1, 2, …, or 16); C_k_ = ~ N (0, σ^2^_cow_); and e_ijk_ = residual error, e_ijk_ ~ N (0, σ^2^_e_).

The average values were used in the statistical analyses. The main effect of the FC ratio, the main effect of MR supplementation, and the interaction between FC and MR were declared significant at *p* ≤ 0.05. When the overall effect of treatment was significant, pairwise differences among means were explored further using Tukey’s test.

## 3. Results

### 3.1. Diet Characteristics

Table 1 presents the ingredients and chemical composition of the experimental diets. The main characteristics of the feeds used to formulate the diets are given in Table 2.

Complete diets (Table 1) for Jersey cows, with 16% CP, were formulated assuming an average production of 26.0 kg milk/d (of diet DM). Chemical analyzes confirmed average values of 16.12–16.19% CP. The energy value of the diets estimated based on the NRC [27] equations was 1.60–1.67 Mcal/kg NE_L_ in the case of LC diets, and 1.67–1.73 Mcal/kg NE_L_ in the case of HC diets, respectively; with higher values being recorded for the diets supplemented with MR (LR and HR). As expected, increasing the proportion of concentrates caused an increase in the starch content of the diets and a decrease in the NDF and ADF content, while supplementing the diets with MR increased the fat content and decreased the starch concentration, which justifies the differences in the content of NE_L_ of the experimental diets.

The MR-supplemented diets contained approximately two-times more FAs, resulting in 52.87–57.86 g total FAs/kg of DM (Table 1). The analysis of the FA composition of the forages indicated that the grass silage contained relatively high proportions of ALA (54.27% of total FAs); while the palmitic acid (PA, C16:0) was predominant in alfalfa hay, and OA was predominant in rapeseed. The concentrates were rich in LA and OA, the latter being predominant in the concentrates that contained rapeseed (LR and HR) (Table 2). Consequently, ALA was the predominant FA in the high FC diets (LC; 35% concentrate), while the low FC diets (HC; 50% concentrate) had a content higher in LA, and diets supplemented with MR (LR and HR diets) were richer in OA (Table 1).

The α-tocopherol content was higher in forages (grass silage: 49.09 mg/kg and alfalfa hay: 32.4 mg/kg of DM) and rapeseed (47.07 mg/kg of DM) compared to concentrates (Table 2) and showed a slight increase with the inclusion of MR in the diets (Table 1). γ-tocopherol did not present important differences between the diets, even if the rapeseeds had a distinctly higher content of this bioactive compound. The content of all-*trans* β-carotenes was increased in grass silage and alfalfa hay (Table 2) as well as α-tocopherol, which determined a higher level of these antioxidants in the high FC diets (LC diets) (Table 1).

### 3.2. Feed Intake, Milk Yield and Milk Composition

Increasing the proportion of concentrates in the feed of cows led to an increase (*p* < 0.01) of DM intake (DMI), an increase (*p* ˂ 0.001) of starch intake, and of NE_L_ (Mcal/d), but had no effect (*p* > 0.05) on the intake of fat, NDF or ADF. The decrease in FC ratio increased (*p* < 0.01) the daily intake of total FAs, PA, stearic acid (SA, C18:0), OA, LA, but did not influence (*p* > 0.05) the intake of ALA (Table 3).

Supplementing the diets with MR had no effect on DMI, but caused an increase in NE_L_ intake (Mcal/d) following an increase (*p* ˂ 0.001) in fat intake, which practically doubled (Table 3). Compared to the diets not supplemented with MR (L and H), the intake of total FAs, OA, LA, and ALA was higher (*p* ˂ 0.01) in cows fed diets supplemented with MR (LR and HR) (Table 3).

HC diets increased milk production (including ECM and 4%FCM) and milk protein content but decreased fat and milk urea nitrogen content compared to LC (Table 3). Supplementing diets with MR increased milk production without affecting the composition of the milk (Table 3). However, fat production, milk lactose percentage and yield, and milk solids-not-fat (SNF) concentrations were not affected by the experimental treatments.

The best feed conversion rate was obtained in the case of cows fed with LC diets, where the consumption of concentrates (g DM/kg milk) was lower by up to 38.5% (*p* ˂ 0.001), compared to that obtained in the case of HC diets. Energy efficiency (Mcal/kg milk) was not affected by the experimental treatments. In addition, supplementing the diets with MR did not affect the consumption of concentrates (g/kg milk) nor the energy efficiency (Mcal/kg milk) of the feed (Table 3).

### 3.3. Milk FAs Profile

Table 4, Table 5 and Table 6 present the effects of different FC ratios of diets with or without MR supplements, used in the feed of Jersey cows, on the composition of FAs and health lipid indices in milk.

Milk FAs composition was modified by both FC ratio and MR supplementation (Table 4). Decreasing the FC ratio of the diets led to a decrease in the concentration of SFAs and an increase in MUFAs and PUFAs in milk fat, while supplementing diets with MR reduced (*p* ˂ 0.001) the proportion of SFAs and increased (*p* ˂ 0.001) the level of MUFAs in milk fat (Table 5).

The concentration of PA and SA was higher (*p* < 0.01) in cows fed LC diets compared to cows fed HC diets, while supplementing MR diets had the effect of decreasing the level of PA and increasing the content of SA in milk (Table 4). However, the increase of SA in milk was higher when the diet was rich in concentrates and supplemented with MR, as shown by the interaction between FC × MR supplementation.

The concentration of OA was not influenced by the FC ratio, instead it increased significantly (*p* ˂ 0.01) when the diets were supplemented with MR. The content of milk in LA and CLA (especially the *cis*-9, *trans*-11 CLA isomer) was higher (*p* ˂ 0.01) in cows fed the diets rich in concentrates (HC diets), while the concentration of ALA was higher (*p* ˂ 0.001) in cows fed with diets rich in forage (LC diets) (Table 4). The supplementation of diets with MR caused a decrease (*p* ˂ 0.01) in LA content and an increase (*p* ˂ 0.05) in total C18:1, VA, and ALA in milk, while the concentration of CLA did not change. FC × MR interaction suggested a more pronounced increase (*p* < 0.01) in the concentration of total C18:1, OA, and VA and a more pronounced decrease in the level of LA in milk fat, when diets rich in concentrates were supplemented with MR (Table 4).

Feeding Jersey cows with diets rich in concentrates (HC—50% concentrate of DM) caused a decrease (*p* ˂ 0.01) in the amount of PA, SA, and n-3 FA and a significant increase in the concentrations of VA, LA, and *cis*-9, *trans*-11 CLA, compared to diets with low concentrate content (LC—35% concentrate of DM). Supplementation of MR (LR and HR diets) significantly decreased PA and increased C18 concentrations in milk, except LA and *cis*-9, *trans*-11 CLA, compared to diets without MR.

Diets with low FC and those containing MR, irrespective of the FC ratio, caused the decrease in the concentration of FAs with a hypercholesterolemic effect (HFA: C12:0 + C14:0 + C16:0) and the increase in the concentration of FAs with a hypocholesterolemic effect (hFA: *cis*-C18:1 total and PUFAs) and the amount of *cis*-type FAs (Table 5).

Increasing the proportion of concentrates in the feed resulted in increasing the concentration of n-6 FAs in milk and supplementing the feed with MR significantly reduced the content of n-6 FAs and increased the level of n-3 FAs, so that the ratio n-6/n-3 FAs became more favorable to human health compared to diets not supplemented with MR (Table 6). Regarding the effects of the FC ratio on health-related lipid indices, the AI and TI decreased, while the h/H FAs ratio increased in the milk of cows fed the HC diets, but also when the diets were supplemented with MR, regardless of the FC ratio of the diet. Increasing the amount of concentrates, and also supplementing the diets with MR, improved the quality of milk fats, an aspect supported by the increase in HPI values (Table 6).

### 3.4. Fat-Soluble Vitamins and Antioxidant Capacity

Table 7 shows the effects of the experimental diets on the content of α-tocopherol, retinol and β-carotenes in raw and pasteurized milk stored for a period of 4 days in the refrigerator, as well as its TAC.

The concentrations of non-enzymatic lipophilic antioxidants in milk were influenced by the experimental treatments but also by the storage time.

α-tocopherol, retinol and all-*trans* β-carotenes showed a higher concentration (*p* ˂ 0.05) in raw milk when cows’ diet had a low content of concentrates (LC diets) compared to diets rich in concentrates (HC diets). The supplementation of diets with MR determined an increase (*p* ˂ 0.05) in the content of α-tocopherol in milk, while retinol and all-*trans* β-carotenes showed a tendency to decrease, even if this was not statistically ensured. Storing pasteurized milk for 4 days in the refrigerator had a negative effect on the level of lipophilic antioxidants, but the differences were significant only in the case of retinol concentrations and the level of all-*trans* β-carotenes in the milk of cows fed with LC diets.

Increasing the amount of concentrates in the feed reduced the TAC value, while supplementing diets with MR improved (*p* < 0.05) the antioxidant capacity of both raw milk and milk stored for 4 days in the refrigerator (Table 7). The highest TAC value was recorded for raw milk obtained from cows fed with the LR diet (with a high FC ratio and supplemented with MR). Storing milk for 4 days in the refrigerator caused a decrease in the total antioxidant capacity in all experimental treatments, the decreases being significant only in the case of HC diets (Table 7), probably due to the higher content of UFAs (Table 5). On the other hand, the lowest antioxidant capacity was recorded in the milk obtained from cows fed the H diet (with low FC ratio and no MR) after 4 days of storage.

## 4. Discussion

The aim of this research was to evaluate the effect of diets with different FC ratios, supplemented or not with MR, on the intake, production and composition of milk, the FAs profile of fat, and the TAC of milk.

In general, DMI in ruminants increases with the decreasing NDF content of the diet [18,41]. In this research, decreasing the FC ratio increased the DMI, which is in agreement with a lower NDF content of the concentrate-rich diets. Previous studies demonstrated that DMI was strongly correlated with NE_L_ intake in dairy cows (r = 0.98), this conclusion being confirmed by the present study [41].

Fats, especially those rich in UFAs, can reduce feed digestibility, can modify rumen fermentation processes (acetic acid to propionic acid ratio), ultimately leading to a decrease in DMI, especially when fats exceed 6–7% of DM [27]. In this research, supplementing the feed with MR even if it led to an increase in the fat content to approximately 6% of DM, the intake of DM was not affected. Similar results were obtained by Chichlowski et al. [42] when the feed of dairy cows was supplemented with ground rapeseed seeds, which increased the fat concentration in the diet to 6.4% (of DM), concluding that this amount of fat is not sufficient to change the rumen pH, acetate-propionate ratios in the rumen, and consequently feed consumption. In addition, Egger et al. [43] and Kliem et al. [44] found that DMI did not change when the diet of dairy cows was supplemented with 73–96 g rapeseed/kg DM.

Milk production was influenced both by the FC ratio and by supplementing the diets with MR, via their effect on DMI and NE_L_ intake (Table 3). The increase in milk production in cows fed with diets rich in concentrates (HC) is due to the higher intake of easily fermentable proteins and carbohydrates, but also to the lower content of NDF and ADF, which favors the production of propionate and the synthesis of microbial proteins in the rumen [17,18,45].

In a meta-analysis by Plata-Pérez et al. [1], it was found that supplementing the diet of cows with oilseeds can increase milk production due to the increase in NE_L_ intake. In our study, the increased milk production in cows fed the HR diet compared to the H diet, was consistent with the higher energy density of the HR diet and, respectively, the higher intake of NE_L_ (Table 3). Even if MR supplementation of the diet that had a low proportion of concentrates led to an increase in the energy density of the feed (LR compared to L), still, the daily milk production did not increase, probably due to the change in rumen fermentation and the decrease in NDF digestibility and ADF in the presence of supplemented fats [6,46].

The reduction of milk fat concentration in cows fed diets rich in concentrates (HC diets) is probably due to the change in rumen fermentation and the formation of *trans* FAs, which inhibits FAs synthesis in the mammary gland, an aspect supported by the significant increase in total *trans* C18:1 in milk fat (Table 4) [23]. The *trans* double bond of *trans* type FAs is formed during rumen fermentation and their proportion in milk fat increases in cows fed diets rich in UFAs, without leading to a decrease in milk fat percentage [42,47]. On the other hand, diets rich in concentrates increase propionate production in the rumen and reduce the amount of precursors of the de novo synthesis of milk FAs (i.e., acetate and butyrate), thus reducing milk fat concentrations [7,48].

This research confirms the findings reported by Nielsen et al. [49] suggesting that an increased proportion of C18:1 *trans*-10 in milk fat is determined by a diet high in starch and low in NDF, which are associated with a reduction in milk fat content. Moreover, Lock et al. [50] associated the high level of C18:1 *trans*-10 in milk with the decrease in milk fat concentration.

Milk protein content was higher in diets with a low FC ratio, probably due to the relationship between milk protein content and the provision of glucogenic nutrients (the relationship between dietary starch content and protein content from milk: R^2^ = 0.40; [45]).

Some researchers [51] reported a decrease in milk fat content, while others reported that dietary fat supplementation had no effect [6,11]. When the diet is high in fat, the decrease in milk fat percentage occurs only when the *trans*-10 C18:1 isomer has a higher concentration compared to the *trans*-11 C18:1 isomer [47]. In this research, supplementing the diet with MR did not lead to a decrease in the milk fat percentage, and the proportion of the *trans*-10 C18:1 isomer in the milk fat was much lower compared to the *trans*-11 C18:1 isomer (Table 4), thus confirming the results of previous studies [42,47]. In addition, it was found that oilseeds that have a high content of LA (e.g., sunflower seeds, cotton seeds, soybeans) induce a significant decrease in milk fat concentration, because LA has an effect of inhibiting the fermentation processes in the rumen, and it is believed that LA is a substrate in the production of *trans* isomers, which are related to the milk fat depression syndrome [7,47]. Flax seeds and rapeseed seeds, having a lower content of LA, form a smaller number of isomers that inhibit the de novo synthesis of FAs with short and medium chains in the mammary gland and thus have a smaller influence on the milk fat concentration [1].

A number of studies have reported a close inverse relationship between the concentration of *cis*-9, *trans*-11 CLA in milk and milk fat content [19,22,52]. In this research, the decrease in the FC ratio reduced the milk fat content but significantly increased the concentration of *cis*-9, *trans*-11 CLA, while supplementing diets with MR had no effect on these parameters (Table 3 and Table 4).

The introduction of vegetable oils in the feed of cows affects the synthesis of microbial proteins, resulting in a decrease in the intake of amino acids in the duodenum and, respectively, in the concentration of milk proteins. It has been estimated that for every 100 g of fat consumed, milk protein concentration decreases by 0.03 percentage points [7,47]. Therefore, this research is in agreement with those previously reported, but the decrease in milk protein concentration as a result of supplementing the diets with MR was smaller: per 100 g of supplemented fat, the milk protein concentration decreased by 0.02 percentage points. It is possible that in this research the relative decrease of ruminal microorganisms, caused by the intake of PUFAs in the rumen through rapeseed, is lower, and thus the intake and absorption of amino acids in the duodenum is higher than in the case of other oleaginous seeds richer in PUFAs.

The larger amounts of concentrates in the cows’ feed favor the synthesis of propionic acid in the rumen, which in intermediate metabolism leads to a larger amount of glucose, and this causes an increase in the concentration of lactose in milk [45]. In this research, the decrease in the FC ratio was not accompanied by an increase (*p* ˃ 0.05) in the concentration of lactose in milk, although a trend in this direction was observed, probably due to the low threshold for increasing the weight of concentrates in the cows’ feed from 35% to 50%.

The inclusion of rapeseed in the diet of cows reduced the content of SFAs in milk, possibly by decreasing the production of acetic acid in the rumen, which is the main substrate for the de novo synthesis of medium-chain FAs [1,15].

In this research, increasing the proportion of concentrates in the cows’ feed, but also supplementing the diets with MR, led to a significant decrease in the concentration of PA in milk, thus confirming the conclusions of previous studies that reported that diets rich in starch and supplemented with vegetable oils inhibit the de novo synthesis of PA in the mammary gland [44]. The lowest concentration of PA was found in the milk of cows fed the high-concentrate (HC) diet and could be attributed to the lower ruminal pH that occurs after the consumption of concentrates, thus confirming the previous observations of Gómez-Cortés et al. [53]. The decrease in milk fat C16:1 *cis*-9 content with the MR treatments is consistent with responses reported earlier [15], where a control diet was compared with diets containing rapeseed oil or rapeseeds milled with wheat. The lower C16:0 and C16:1 *cis*-9 concentrations in milk from the MR treatments was probably due, in part, to the reduction in de novo synthesis brought about by long chain UFAs [15].

This research suggests that supplementing diets with MR favored the biohydrogenation of UFAs from feed, causing a significant increase (*p* < 0.001) in the level of SA in milk fat [54,55,56]. The increase in the concentration of OA in milk, in the case of diets supplemented with MR, was due to the fact that the daily intake of OA of cows fed with diets supplemented with MR (LR and HR) was higher (157 and 235 g/d, respectively) compared to the diet without MR (L and H) (41.9 and 86.5 g/d, respectively) (Table 3). Vogdanou [56] considers that the increased concentration of OA in milk is due to the fact that this FAs, present in large quantities in rapeseed, was not completely hydrogenated in the rumen to SA, and a small amount was isomerized and transformed into VA. The increase in the concentration of OA in milk fat can be determined by the increased dietary intake of OA from rapeseed but also by the improvement of desaturase activity in the mammary gland [53]. In this research, desaturase activity decreased (*p* ˂ 0.01) by supplementing the diet with MR, thus not confirming the conclusions formulated by Gómez-Cortés et al. [53].

Larsen et al. [55] reported a close relationship between the concentration of VA and *cis*-9, *trans*-11 CLA in milk, as the latter is mainly formed by the desaturation of VA in the mammary gland in the presence of Δ^9^-desaturase. In this research, VA increased by supplementing diets with MR (LR and HR diets compared to L and H diets), but the concentration of *cis*-9, *trans*-11 CLA in milk was not affected (Table 4). The results similar to those obtained by us were previously reported by Egger et al. [43] and Suli et al. [3], who concluded that the inclusion of rapeseed or flaxseed in the diet of cows did not influence the content of *cis*-9, *trans*-11 CLA in milk fat. Other studies have shown that vegetable oil supplementation can influence the concentration of *cis*-9, *trans*-11 CLA in milk only when high amounts of oil are used in a diet that has an extreme FC ratio [22].

Djordjevic et al. [57] reported that OA may be an important precursor for the biohydrogenation of VA. In this research, this conclusion was not confirmed, because the supplementation of diets with MR, which have a high content of OA, did not increase the concentration of *cis*-9, *trans*-11 CLA in milk, which means that the intake increased OA in the feed did not increase the concentration of VA in the rumen. There is research that showed that supplementing the diet with vegetable oils increased the concentration of *cis*-9, *trans*-11 CLA in milk fat in the first week and then decreased and remained relatively constant [9,20,58]. In this research, the determination of the FAs profile in milk was conducted in the third week after the start of supplementing the diet with MR.

The higher amount of concentrates in the feed led to a higher concentration of VA in the rumen and a higher level of *cis*-9, *trans*-11 CLA in milk (Table 4). The increase in the concentration of VA in milk is directly related to the increase in the concentration of VA in the rumen, but also with a higher proportion of *cis*-9, *trans*-11 CLA in milk [8,59]. In this research, this theory was confirmed by the intensification of desaturation processes at the level of the mammary gland, where a part of VA in the presence of Δ^9^-desaturase was transformed into *cis*-9, *trans*-11 CLA (Table 4). In addition, the diets rich in concentrates (HC diets) ensured a higher intake of LA which through rumen biohydrogenation leads to the synthesis of VA from which CLA is later synthesized [8,60].

The introduction of rapeseed (6.4% of DM diet) in diets with different FC ratios determined an increase in the concentration of total C18:1 in milk by 12.8–23.24% (*p* ˂ 0.001; Table 4). The results of this research are comparable to those obtained by Chichlowski et al. [42] when they supplemented the diet of cows with rapeseed seeds (14% of DM diet); but Aldrich et al. [61] obtained a much more pronounced increase in the concentration of C18:1 in milk (by 67%) by supplementing the diet with ground rapeseed (11.2% of the DM diet). The differences in C18:1 concentration in milk fat in the current study could be attributed to the lower proportion of concentrates in the feed, the lower level of rapeseed seed supplementation, or both, compared to the study by Aldrich et al. [61].

It is known that increasing the proportion of concentrates in feed favors the formation of *trans*-10 C18:1 [1,62], and MR contains oleic acid which is an important precursor for *trans*-10 C18:1 in the rumen. The concentration of other *trans* C18:1 isomers in milk, such as *trans* (6+7+8) and *trans*-9 C18:1, increased with the increase in the proportion of concentrates, but also by supplementing the diets with MR (*p* < 0.01; Table 4). These increases of some C18:1 *trans* isomers in milk are in agreement with the studies carried out by Matamoros et al. [62], who concluded that these FAs come from the biohydrogenation of OA in the rumen, their percentages in milk depend on the proportion of *cis*-9 C18:1 in the diet. The results obtained in this study confirm that the interaction between the FC ratio of the diets, and their supplementation with MR has a major importance in determining the total *trans* C18:1 concentrations in milk fat.

Supplementing the diet with MR led to an increase in the concentration of ALA (*p* ˂ 0.05) in milk fat, and to a decrease in the level of LA (*p* ˂ 0.01), while the increase in the proportion of concentrates in the cows’ ration had the opposite effect, namely the concentration of LA increased (*p* ˂ 0.01) and the level of ALA decreased (*p* ˂ 0.001) (Table 4); this being in agreement with the concentration of these FAs in the cows’ diet (Table 1). In fact, diets supplemented with MR provided more than twice the amount of ALA than the diets not supplemented with MR, in terms of intake, regardless of the proportion of concentrates in the feed (297.8 vs. 127.7 g/d for diets LC and 292.3 vs. 137.3 g/d for HC diets, respectively) (Table 3). These aspects influenced the n-6/n-3 FAs ratio, which decreased (*p* ˂ 0.001) by supplementing the diet with MR and increased (*p* ˂ 0.001) by increasing the proportion of concentrates in the feed.

The tendency to decrease the proportion of PUFAs in milk when the diets were supplemented with MR suggests that a high proportion of rapeseed fat was saturated by the rumen biohydrogenation of UFAs, this hypothesis being supported by the increase in the concentration of VA and the decrease in the level of LA (Table 4). Increasing the concentration of VA in milk fat is beneficial for consumers, because in the human body, VA under the action of Δ^9^-desaturase is transformed into *cis*-9, *trans*-11 CLA [4,47].

Often, the assessment of the quality of milk fats in terms of the impact on human health is based on the ratio between certain FAs groups (PUFA/SFA, n-6/n-3, h/H), the AI, the TI, and the HPI. In the present study, the PUFA/SFA ratio was considerably lower (0.07–0.094) than the recommended values for human nutrition (over 0.45), in all experimental treatments, due to the high concentration of SFAs in milk [5,10,63]. An n-6/n-3 FAs ratio of less than 4 is recommended in human nutrition as a means of preventing cardiovascular diseases and cancer [5,38]. Thus, considering the highest n-3 FAs content and the lowest n-6/n-3 FAs ratio in milk fat, the diet with a high FC ratio (65:35) supplemented with MR seems to increase the nutritional value of milk.

Milk that has a low AI and TI value, and higher h/H FA and HPI values, has a lower risk of contributing to the development of cardiovascular diseases, and thus can be considered a functional food with benefits for human health [5,38]. Although no organization has yet established recommended values for these fat quality indices [4,64], in this research we found that milk with the best fat quality was obtained from cows fed a diet with low FC and supplemented with MR (HR diet), while the diet with high FC and no MR supplement (L diet) determined the lowest quality of milk fat, in terms of impact on human health.

It is extremely important to determine the antioxidant capacity of milk because oxidation can lead to the deterioration of the nutritional quality of milk and the appearance of unpleasant flavors. The oxidative stability of milk depends mainly on the composition of FAs, as pro-oxidant factors, and on the concentration of tocopherols and carotenoids as antioxidant factors [2,24].

The concentration of lipophilic antioxidants in milk is directly influenced by the level of fat-soluble vitamins in animal feed [65]. In this research, the higher concentrations of lipophilic antioxidants (α-tocopherol, retinol and all-*trans* β-carotenes), recorded in the milk of cows fed with LC diets (Table 7), are due to the significantly higher intake of α-tocopherol and all-*trans* β-carotenes through food (Table 3). On the other hand, grass silage and alfalfa hay containing 5–10 times more carotenoids than concentrates [66] can justify the higher concentration (*p* ˂ 0.05) of retinol and all-*trans* β-carotenes in the milk of LC cows compared to HC. The concentration of retinol in milk, observed in this research in cows fed diets not supplemented with MR, is comparable to that reported by Mogensen et al. [67]; however, the level of α-tocopherol and all-*trans* β-carotenes was up to two times higher in this research, probably due to the higher dietary intake of antioxidants.

As expected, supplementing diets with rapeseed increased (*p* ˂ 0.05) the concentration of α-tocopherol in milk (Table 7). The research carried out by Vogdanou [56] and Larsen et al. [55] showed a significant increase in the amount of α-tocopherol in milk when the cows’ diet was supplemented with oilseeds (rape, flax) or rapeseed cake, concluding that the concentrations of tocopherols can be increased by increasing their supply from feed.

Noteworthy is the decrease in the retinol content of milk (raw and stored), both as a result of the increase in the amount of concentrates, and as a result of supplementing the feed with MR (Table 7). Studies by Puppel et al. [26] demonstrated that the use of flax seeds in cow feed caused an increase in the concentration of retinol and β-carotene in milk, aspects that were not confirmed in this research by using rapeseed.

Lipid oxidation is an inevitable process in which the double bonds of UFAs are attacked by oxidizing compounds, leading to the degradation of milk [12,68]. Although several authors have shown that milk becomes more susceptible to oxidation when the UFAs content increases, Salles et al. [68] concluded that the oxidative stability of milk is more dependent on the antioxidant content than on the UFAs concentration of fat. In this research, even if supplementing diets with MR led to an increase in the concentration of UFAs in milk, the oxidative stability of milk determined by TAC (µmol TE/mL) actually increased both for raw milk and for pasteurized and stored milk. This is due to the increased intake of antioxidants (especially tocopherols) via MR supplementation, which was reflected in a higher concentration of α-tocopherol in milk, thus inhibiting the oxidation of UFAs in milk.

The supplementation of diets with MR led to increased TAC values in milk samples (*p* < 0.05), the increase being greater in the case of diets with high FC ratio (Table 7). This increase in the TAC of milk was due to the increased concentration of tocopherols in rapeseed, but also to the high content of α-tocopherol and all-*trans* β-carotenes of grass silage, which provides better protection of UFAs and especially to the polyunsaturated ones that are more susceptible to oxidation. An increase in the TAC of milk was observed when the diet was supplemented with flax seeds in cows [26], or with camelina seeds in sheep [69].

The highest antioxidant activity of milk (3.18 µM TE/mL) was associated with the highest concentration of α-tocopherol (1.982 mg/L) measured in raw milk produced by cows fed the LR diet (high FC and supplemented with MR). These results agree with the conclusions of Puppel et al. [26] who mention that with the increase in the level of tocopherols, the antioxidant activity of milk also increases.

In agreement with this research, the studies carried out by El-Fattah et al. [70] demonstrated that pasteurization had no effect on the antioxidant capacity of milk. The authors mention that the antioxidant capacity of milk increases if the temperature during the heat treatment is over 100 °C, due to the release of thiol groups from proteins, which act as hydrogen donors. On the other hand, Sanlidere et al. [71] showed that there are no significant differences between raw, pasteurized, and sterilized milk in terms of antioxidant activity (ABTS), which was 4.02, 4.47, and 4.18 µM TE/mL, respectively.

The storage of milk for 4 days in the refrigerator resulted in a decrease in the concentration of retinol and all-*trans* β-carotenes, probably due to the reactive oxygen species formed during storage which were inactivated by the antioxidants in the milk [72]. As a result of this activity, the antioxidants in milk are oxidized, thus leading to a decrease in their concentration in milk, even α-tocopherol, although this decrease was not statistically significant (Table 7).

During storage, the TAC of milk decreases significantly in the HC groups and does not change significantly, it tends to decrease in the LC groups, being correlated with the concentration of antioxidants in the milk. Havemose et al. [25] demonstrated that high levels of α-tocopherol and retinol in milk do not prevent the oxidation of UFAs but delay this process, thus extending the shelf life of milk and dairy products. Therefore, consumers could benefit from the nutritional qualities and bioactive compounds of milk during the 4 days of keeping the milk in the refrigerator.

## 5. Conclusions

The results obtained in this study demonstrated that the FC ratio of Jersey cow diets affected milk production and composition, while the supplement mentation of diets with MR mainly influenced the milk FAs profile and antioxidant capacity of milk. Increasing the proportion of concentrates in the cows’ diets increased DMI, milk production, and milk protein content, while milk fat concentration decreased. The cows fed diets rich in forages (high FC) produced less milk, but the milk fat had a higher content of n-3 FAs and antioxidants (α-tocopherol, retinol, and all-trans β-carotenes) compared to the diets rich in concentrates (low FC). Data indicated that ground rapeseeds can be used as a part of an overall strategy for reducing milk SFA and increasing concentration of FAs beneficial for human health (OA, VA and ALA), but also for improvement in the concentration of lipophilic antioxidants and TAC.

It is important to continue research to establish the effects of supplementing cow diets with rapeseed, on milk processing, and the properties of dairy products. Increasing the concentration of UFAs and especially OA in milk fat could affect the activity of the starter cultures when making cheese, or could produce a softer butter as a result of the increase in the ratio between OA and PA.

## Figures and Tables

**Figure 1 life-13-00046-f001:**
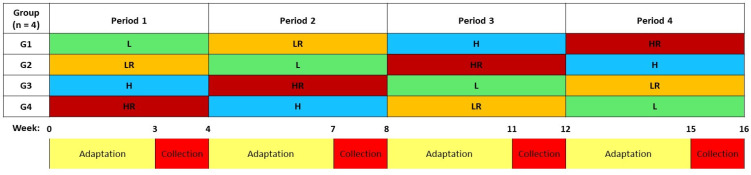
Study design. L—low concentrate (FC ratio 65:35) and no MR; LR—low concentrate (FC ratio 65:35) supplemented with MR; H—high concentrate (FC ratio 50:50) and no MR; HR—high concentrate (FC ratio 50:50) supplemented with MR; MR—milled rapeseed seeds, FC—forage:concentrate; G—group; Adaptation period (3 weeks each): adaptation of animals to the different experimental diets. Collection periods (1 week each): data collection and sampling.Numbers in the timeline indicate weeks of the trial.

**Table 1 life-13-00046-t001:** Ingredients, chemical composition, fatty acid profile, and fat-soluble antioxidants content of experimental diets.

Item		Treatment			
		65:35 (Forage:Concentrate) (LC)		50:50 (Forage:Concentrate) (HC)	
		No Rapeseed (L)	Rapeseed (LR)	No Rapeseed (H)	Rapeseed (HR)
**Ingredient** (DM basis)					
Grass silage		43.3	43.3	33.3	33.3
Alfalfa hay		21.7	21.7	16.7	16.7
Concentrate		35.0	35.0	50.0	50.0
	Rapeseed	-	18.3	-	12.8
	Maize grain	50.8	39.5	56.8	48.7
	Triticale grain	10.0	10.0	10.0	10.0
	Soybean meal	18.0	18.0	12.0	12.0
	Rapeseed meal	15.0	8.0	16.5	11.8
	Sodium bicarbonate	0.9	0.9	0.7	0.7
	Salt	1.2	1.2	1.0	1.0
	Premix mineral and vitaminic	4.1	4.1	3.0	3.0
**Chemical composition**					
DM (%)		57.08 ± 1.7	57.31 ± 2.3	62.30 ± 2.0	62.07 ± 1.5
Nutrients, % of DM					
	Ash	6.9 ± 0.3	7.1 ± 0.3	7.0 ± 0.5	7.2 ± 0.4
	CP	16.12 ± 0.7	16.13 ± 0.5	16.17 ± 0.7	16.19 ± 0.9
	CF	2.95 ± 0.17	5.84 ± 0.30	3.02 ± 0.21	6.03 ± 0.38
	NDF	37.70 ± 1.96	37.92 ± 1.88	32.15 ± 0.92	32.38 ± 1.45
	ADF	22.26 ± 0.44	22.52 ± 0.63	18.58 ± 0.65	18.85 ± 0.46
	Starch	17.81 ± 1.32	14.90 ± 0.90	25.65 ± 1.22	22.71 ± 0.83
NE_L_, Mcal/kg of DM		1.60	1.67	1.67	1.73
**Fatty acids composition**					
Total FAs (g/kg of DM)		24.92 ± 2.3	52.87 ± 1.27	30.34 ± 1.14	57.86 ± 2.18
Fatty acids profile (% of total FAs)					
	C16:0	25.26 ± 0.61	20.41 ± 0.50	25.64 ± 0.93	20.79 ± 0.41
	C18:0	3.73 ± 0.14	3.78 ± 0.10	3.50 ± 0.08	3.43 ± 0.09
	C18:1 *cis*-9 (OA)	9.42 ± 0.34	16.14 ± 0.92	14.64 ± 0.37	20.47 ± 0.73
	C18:2 n-6 (LA)	23.31 ± 1.44	20.26 ± 0.58	23.15 ± 0.90	20.36 ± 0.55
	C18:3 n-3 (ALA)	28.69 ± 2.36	30.57 ± 1.47	23.23 ± 1.78	25.35 ± 1.09
	Other	9.59 ± 0.50	8.83 ± 0.41	10.05 ± 0.54	9.81 ± 0.77
**Fat-soluble antioxidants** (mg/kg of DM)					
α-tocopherol		29.14 ± 3.16	32.27 ± 2.54	23.36 ± 2.75	26.35 ± 1.88
γ-tocopherol		5.38 ± 0.52	6.78 ± 0.49	6.50 ± 0.42	6.78 ± 0.82
All-trans β-carotenes		25.58 ± 1.78	26.07 ± 1.37	20.17 ± 1.20	20.10 ± 0.94

DM—dry matter; CP—crude protein; CF—crude fat; NDF—neutral detergent fiber; ADF—acid detergent fiber; OA—oleic acid; LA—linoleic acid; ALA—α-linolenic acid.

**Table 2 life-13-00046-t002:** Chemical composition, fatty acid profile, and fat-soluble antioxidants content of feedstuffs.

Item		Grass Silage	Alfalfa Hay	Rapeseed	Concentrates ^†^			
					L	LR	H	HR
**Chemical composition**								
DM (%)		38.73	89.30	89.17	88.71	89.17	89.70	88.93
Nutrients, % of DM								
	Ash	9.15	10.17	4.05	6.59	6.32	6.18	6.41
	CP	12.45	16.61	20.25	20.36	20.39	18.50	18.55
	CF	3.30	1.27	47.02	3.56	11.82	3.41	9.44
	NDF	46.20	59.82	19.34	13.47	14.08	13.57	14.02
	ADF	29.86	32.64	13.61	6.43	7.16	6.39	6.92
	Starch	3.42	ND	ND	46.32	38.01	48.85	42.96
NE_L_, Mcal/kg of DM		1.59	1.16	3.08	1.89	2.10	1.90	2.02
**Fatty acids composition**								
Total FAs (g/kg of DM)		21.42	14.70	437.41	35.60	115.48	41.51	96.56
Fatty acids profile (% of total FAs)								
	C16:0	20.07	41.35	4.65	21.70	7.86	24.09	14.41
	C18:0	2.43	8.19	1.91	2.56	2.71	2.65	2.50
	C18:1 *cis*-9 (OA)	2.61	4.12	52.40	21.15	40.35	25.77	37.42
	C18:2 n-6 (LA)	12.24	18.25	19.31	40.14	31.41	32.04	26.47
	C18:3 n-3 (ALA)	54.27	17.95	11.43	3.72	9.08	4.33	8.56
	Other	8.38	10.15	10.30	10.73	8.59	11.12	10.64
**Fat-soluble antioxidants** (mg/kg of DM)								
α-tocopherol		49.09	32.40	47.07	1.62	9.98	3.40	9.18
γ-tocopherol		8.83	3.10	20.75	2.72	6.27	6.31	8.72
All-*trans* β-carotenes		44.19	30.30	1.56	0.31	0.52	0.49	0.67

^†^ L, LR, H and HR refer to concentrates offered in diets with low or high amounts of concentrates without or with the addition of MR (L—low concentrate, without addition of MR; LR—low concentrate, with addition of MR; H—high concentrate, without addition of MR; HR—high concentrate, with addition of MR). DM—dry matter; CP—crude protein; CF—crude fat; NDF—neutral detergent fiber; ADF—acid detergent fiber; OA—oleic acid; LA—linoleic acid; ALA—α-linolenic acid. ND = not determined.

**Table 3 life-13-00046-t003:** Effects of forage to concentrate ratio (FC) and dietary milled rapeseed (MR) supplement on performance of dairy cows.

Item				Treatment ^†^				SEM	*p*-Value ^††^		
				LC		HC					
				L	LR	H	HR		1	2	3
**Intake:**											
	Total DM (kg/d)			17.86	18.42	19.47	19.93	0.64	**	ns	ns
	Grass silage (kg DM/d)			7.73	7.97	6.48	6.64	0.28	**	ns	ns
	Alfalfa hay (kg DM/d)			3.87	4.00	3.52	3.32	0.17	*	ns	ns
	Concentrate (kg DM/d)			6.26	6.45	9.47	9.97	0.08	***	ns	ns
	CP (kg/d)			2.88	2.97	3.15	3.23	0.09	ns	ns	ns
	CF(kg/d)			0.53	1.08	0.58	1.20	0.02	ns	***	ns
	NDF (kg/d)			6.73	6.98	6.26	6.45	0.24	ns	ns	ns
	ADF (kg/d)			3.97	4.15	3.62	3.76	0.13	ns	ns	ns
	Starch (kg/d)			3.18	2.74	4.99	4.53	0.17	***	*	ns
	α-tocopherol (mg/d)			520.4	594.4	454.8	525.1	38.75	**	**	ns
	γ-tocopherol (mg/d)			113.9	124.9	126.6	135.1	8.39	ns	ns	ns
	All-*trans* β-carotenes (mg/d)			456.8	480.2	392.7	400.6	22.63	*	ns	ns
	NE_L_ intake (Mcal/d)			28.58	30.77	32.52	34.48	0.63	***	**	ns
	Fatty acids intake (g/d)										
			Total fatty acids	445	974	591	1153	34.92	**	***	ns
			C16:0	112.4	198.8	151.5	239.7	9.71	**	***	ns
			C18:0	16.6	36.8	20.7	39.6	1.09	**	***	ns
			C18:1 *cis*-9 (OA)	41.9	157.2	86.5	236.0	8.75	**	***	ns
			C18:2 n-6 (LA)	103.7	197.3	136.8	234.7	9.12	***	***	ns
			C18:3 n-3 (ALA)	127.7	297.8	137.3	292.3	7.85	ns	***	ns
**Milk yield (kg/d)**											
		Milk		21.72	21.67	24.40	26.03	1.97	**	*	*
		ECM		29.33	28.79	32.28	33.52	2.21	*	ns	*
		4% FCM		26.51	26.03	28.57	29.66	2.17	*	ns	*
		Fat		1.188	1.157	1.254	1.283	0.042	ns	ns	ns
		Protein		0.995	0.977	1.169	1.218	0.037	**	ns	ns
		Lactose		1.036	1.051	1.174	1.268	0.103	ns	ns	ns
**Milk composition, %**											
		Fat		5.47	5.34	5.14	4.93	0.11	**	ns	ns
		Protein		4.58	4.51	4.79	4.68	0.07	**	ns	ns
		Lactose		4.77	4.85	4.81	4.87	0.09	ns	ns	ns
		SNF		9.21	9.36	9.08	9.17	0.09	ns	ns	ns
		Urea-N (mg/dL)		24.6	23.4	20.7	20.9	0.87	*	ns	ns
**Feed conversion ratios:**											
	Concentrate (g/kg milk)			288.2	297.6	399.2	383.0	10.41	***	ns	ns
	NE_L_ intake (Mcal/kg milk)			1.32	1.42	1.34	1.33	0.06	ns	ns	ns

^†^ LC—low concentrate (FC = 65:35); HC—high concentrate (FC = 50:50); L—low concentrate, without addition of MR; LR—low concentrate, with addition of MR; H—high concentrate, without addition of MR; HR—high concentrate, with addition of MR. ^††^ 1—forage to concentrate ratio (FC), 2—milled rapeseed (MR), 3—FC × MR. ns: *p* > 0.05; * *p* < 0.05; ** *p* < 0.01 *** *p* < 0.001. DM—dry matter; CP—crude protein; CF—crude fat; NDF—neutral detergent fiber; ADF—acid detergent fiber; OA—oleic acid; LA—linoleic acid; ALA—α-linolenic acid. ECM = Energy Corrected milk; 4% FCM = Fat Corrected Milk. OA—oleic acid; LA—linoleic acid; ALA—α-linolenic acid; SNF: solids non-fat.

**Table 4 life-13-00046-t004:** Effects of forage to concentrate ratio (FC) and dietary milled rapeseed (MR) supplement on milk fatty acids (FAs) composition (% of total FAs).

Item	Treatment ^†^				SEM	*p*-Value ^††^		
	LC		HC					
	L	LR	H	HR		1	2	3
Total FAs (g/100 g fat milk)	94.32	94.61	94.52	94.47	0.097	ns	ns	ns
C4:0	1.10	1.04	0.70	0.67	0.043	**	ns	ns
C6:0	1.36	1.35	1.52	1.57	0.054	ns	ns	ns
C8:0	0.99	1.02	0.89	0.85	0.050	ns	ns	ns
C10:0	2.56	2.52	2.87	2.76	0.084	*	ns	ns
C10:1 *cis*-9	0.43	0.39	0.46	0.41	0.017	ns	ns	ns
C12:0	2.98	3.45	2.65	2.73	0.159	**	*	*
C12:1 *cis*-9	0.14	0.08	0.12	0.07	0.008	ns	*	ns
C13:0	0.20	0.17	0.17	0.18	0.012	*	*	*
C14:0	11.44	12.81	12.12	12.78	0.271	*	ns	ns
C14:1 *cis*-9	1.62	1.96	2.47	2.77	0.069	**	*	ns
C15:0	0.91	0.83	0.86	0.90	0.032	*	ns	ns
C15:1	0.14	0.13	0.12	0.11	0.004	ns	ns	ns
C16:0	31.87	22.98	29.10	18.91	0.671	**	***	**
C16:1 *cis*-9	1.39	1.08	2.17	1.59	0.055	**	**	ns
C17:0	0.50	0.45	0.48	0.53	0.021	ns	ns	ns
C17:1	0.30	0.20	0.18	0.15	0.011	***	**	*
C18:0	10.92	15.96	8.81	14.32	0.378	***	***	**
C18:1 *trans*-(6+7+8)	0.16	0.35	0.21	0.36	0.107	**	**	ns
C18:1 *trans*-9	0.16	0.31	0.20	0.32	0.012	**	**	ns
C18:1 *trans*-10	0.25	0.46	0.38	0.75	0.110	***	***	*
C18:1 *trans*-11 (VA)	1.05	1.53	1.87	2.76	0.240	***	***	**
C18:1 *cis*-9 (OA)	21.51	23.74	22.72	27.32	0.765	ns	**	*
C18:1 *cis*-(12+13)	0.32	0.58	0.37	0.67	0.032	*	**	ns
C18:1 *trans* total	1.62	2.36	2.66	3.95	0.112	***	**	*
C18:1 *cis* total	21.83	24.09	22.59	27.17	0.548	ns	**	*
C18:1 total	23.45	26.45	25.25	31.12	0.753	**	***	**
C18:2 *trans*-9, *trans*-12	0.04	0.06	0.04	0.05	0.003	ns	ns	ns
C18:2 *cis*-9, *cis*-12 (LA)	2.40	2.16	3.20	2.66	0.139	**	**	*
CLA total	0.97	0.94	1.50	1.48	0.087	**	ns	ns
CLA *cis*-9, *trans*-11 (RA)	0.81	0.82	1.37	1.38	0.050	**	ns	ns
CLA *trans*-10, *cis*-12	0.16	0.12	0.13	0.10	0.002	ns	ns	ns
C18:3 *cis*-6, *cis*-9, *cis*-12	0.06	0.04	0.06	0.04	0.002	ns	ns	ns
C18:3 *cis*-9, *cis*-12, *cis*-15 (ALA)	0.71	0.90	0.53	0.67	0.042	***	*	*
C20:0	0.20	0.22	0.19	0.19	0.007	ns	ns	ns
C20:4 n-6 (AA)	0.14	0.13	0.13	0.14	0.001	ns	ns	ns
C20:5 n-3 (EPA)	0.16	0.12	0.06	0.04	0.001	***	*	ns
C22:0	0.11	0.15	0.10	0.12	0.003	ns	ns	ns
C22:4 n-6	0.05	0.04	0.04	0.03	0.006	ns	ns	ns
C22:5 n-3 (DPA)	0.11	0.12	0.10	0.11	0.041	ns	ns	ns
Unidentified fatty acids	2.75	2.23	2.61	2.05	0.112	ns	ns	ns

^†^ LC—low concentrate (FC = 65:35); HC—high concentrate (FC = 50:50); L—low concentrate, without addition of MR; LR—low concentrate, with addition of MR; H—high concentrate, without addition of MR; HR—high concentrate, with addition of MR. ^††^ 1—forage to concentrate ratio (FC), 2—milled rapeseed (MR), 3—FC × MR. ns: *p* > 0.05; * *p* < 0.05; ** *p* < 0.01 *** *p* < 0.001. VA: vaccenic acid; OA: oleic acid; LA: linoleic acid; CLA: conjugated linoleic acid; RA: rumenic acid; ALA: α-linolenic acid; AA: arachidonic acid; EPA: eicosapentaenoic acid; DPA: docosapentaenoic acid.

**Table 5 life-13-00046-t005:** Effects of forage to concentrate ratio (FC) and dietary milled rapeseed (MR) supplement on milk fatty acids sums (% of total FAs).

Item		Treatment ^†^				SEM	*p*-Value ^††^		
		LC		HC					
		L	LR	H	HR		1	2	3
Ʃ SFA		65.14	62.95	60.46	56.51	1.120	***	***	*
Ʃ MUFA		27.47	30.29	31.27	36.22	0.878	***	***	**
Ʃ *trans* total		2.10	2.75	3.00	4.26	0.152	**	*	*
Ʃ *trans* MUFA		2.06	2.69	2.96	4.21	0.098	*	*	*
Ʃ *cis* MUFA		25.41	27.60	28.31	32.01	0.755	***	***	**
Ʃ PUFA		4.57	4.52	5.66	5.22	0.193	**	ns	ns
Ʃ n-6 PUFA ^1^		2.65	2.43	3.43	2.92	0.078	**	*	*
Ʃ n-3 PUFA ^2^		0.98	1.14	0.69	0.82	0.054	***	*	ns
Ʃ UFA		32.04	34.81	36.93	41.44	0.756	**	*	*
Ʃ *trans* FA + SFA		67.24	65.70	63.46	60.77	1.254	**	*	ns
Ʃ *cis* FA ^3^		29.04	31.11	32.43	35.70	0.961	***	**	ns
HFA		46.29	39.24	43.87	34.42	0.864	*	***	*
hFA		28.02	30.97	30.91	36.34	0.721	*	**	*
Product/substrate ratios:									
	*c*9 14:1/*c*9 14:1 + 14:0	0.124	0.132	0.169	0.178	0.011	**	ns	ns
	*c*9 16:1/*c*9 16:1 + 16:0	0.042	0.045	0.069	0.077	0.003	***	*	*
	*c*9 18:1/*c*9 18:1 + 18:0	0.663	0.598	0.721	0.656	0.012	**	**	ns
	*c*9,*t*11 CLA/*c*9,*t*11 CLA + *t*11 18:1	0.435	0.389	0.623	0.573	0.010	*	ns	ns
	Δ^9^-desaturase index	0.314	0.341	0.356	0.404	0.051	*	*	ns

^†^ LC—low concentrate (FC = 65:35); HC—high concentrate (FC = 50:50); L—low concentrate, without addition of MR; LR—low concentrate, with addition of MR; H—high concentrate, without addition of MR; HR—high concentrate, with addition of MR. ^††^ 1—forage to concentrate ratio (FC), 2—milled rapeseed (MR), 3—FC × MR. ns: *p* > 0.05; * *p* < 0.05; ** *p* < 0.01 *** *p* < 0.001. HFA: Hypercholesterolemic FA (C12:0 + C14:0 + C16:0); hFA: hypocholesterolemic FA (C18:1 + PUFA); ^1^ Sum of 18:2 *cis*-12, *cis*-15; 18:3 *cis*-6, *cis*-9, *cis*-12; 20:4 n-6 and 22:4 n-6. ^2^ Sum of 18:3 *cis*-9, *cis*-12, *cis*-15; 20:5 n-3 (EPA); 22:5 n-3 (DPA); ^3^ Sum of *cis*-MUFA and *cis*-PUFA.

**Table 6 life-13-00046-t006:** Effects of forage to concentrate ratio and dietary milled rapeseed supplement on health lipid indices in milk.

Item	Treatment ^†^				SEM	*p*-Value ^††^		
	LC		HC					
	L	LR	H	HR		1	2	3
PUFA/SFA	0.070	0.072	0.094	0.092	0.011	***	ns	ns
HFA/UFA	1.445	1.127	1.188	0.831	0.114	**	**	ns
n-6/n-3 FA	2.70	2.13	4.97	3.56	0.178	***	**	*
AI	2.52	2.23	2.17	1.76	0.106	**	***	*
TI	2.49	2.09	2.07	1.61	0.098	**	**	ns
h/H FA	0.605	0.789	0.705	1.056	0.037	**	***	*
HPI	0.386	0.436	0.441	0.549	0.043	**	**	*

^†^ LC—low concentrate (FC = 65:35); HC—high concentrate (FC = 50:50); L—low concentrate, without addition of MR; LR—low concentrate, with addition of MR; H—high concentrate, without addition of MR; HR—high concentrate, with addition of MR. ^††^ 1—forage to concentrate ratio (FC), 2—milled rapeseed (MR), 3—FC × MR. ns: *p* > 0.05; * *p* < 0.05; ** *p* < 0.01 *** *p* < 0.001. HFA: hypercholesterolaemic FA (12:0 + 14:0 + 16:0); UFA: unsaturated fatty acid; AI: Atherogenicity Index; TI: Thrombogenicity Index; h: hypocholesterolemic FA; H: Hypercholesterolemic FA; HPI: Health Promoting Index.

**Table 7 life-13-00046-t007:** Effects of forage to concentrate ratio and dietary milled rapeseed supplement on antioxidant content and total antioxidant capacity (TAC) of raw (R), pasteurized (P) and stored milk (S).

Item		Treatment ^†^				SEM	*p*-Value ^††^		
		LC		HC					
		L	LR	H	HR		1	2	3
α-tocopherol (mg/L)	R	1.714	1.982	1.434	1.617	0.193	*	*	ns
	S	1.673	1.854	1.412	1.572	0.241	*	*	ns
Retinol (mg/L)	R	0.554 ^a^	0.478 ^a^	0.396 ^a^	0.367 ^a^	0.045	*	ns	ns
	S	0.481 ^b^	0.402 ^b^	0.317 ^b^	0.288 ^b^	0.033	**	ns	ns
All-*trans* β-carotenes (mg/L)	R	0.321 ^a^	0.283 ^a^	0.263	0.225	0.042	*	ns	ns
	S	0.256 ^b^	0.215 ^b^	0.242	0.202	0.040	*	ns	*
TAC (µM TE/mL)	R	2.71	3.18	2.47 ^a^	2.83 ^a^	0.190	*	**	*
	P	2.85	3.34	2.59 ^a^	2.78 ^a^	0.241	*	*	ns
	S	2.60	3.02	2.05 ^b^	2.34 ^b^	0.236	**	*	*

^†^ LC—low concentrate (FC = 65:35); HC—high concentrate (FC = 50:50); L—low concentrate, without addition of MR; LR—low concentrate, with addition of MR; H—high concentrate, without addition of MR; HR—high concentrate, with addition of MR. ^††^ 1—forage to concentrate ratio (FC), 2—milled rapeseed (MR), 3—FC × MR. ns: *p* > 0.05; * *p* < 0.05; ** *p* < 0.01. ^a,b^ Mean values within a column without a common superscript corresponding to a parameter (R vs. S) differ significantly (*p* < 0.05).

## Data Availability

Not applicable.

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
