# Peer review of "The Effects of Feeding Milled Rapeseed Seeds with Different Forage:Concentrate Ratios in Jersey Dairy Cows on Milk Production, Milk Fatty Acid Composition, and Milk Antioxidant Capacity"

_life, 2022, doi:10.3390/life13010046_

Round 1

Reviewer 1 Report

Topic: The Effects of Feeding Milled Rapeseed Seeds with Different 3 Forage: Concentrate Ratios in Jersey Dairy Cows on Milk Production, Milk Fatty Acid Composition, and Milk Antioxidant 5 Capacity.

-The introduction is so many paragraphs, should be 3-4 paragraph will be make it easy to understand.

-Some references have to in update.

Line 193: How to analysis for CP (Like type of machine, where is the machine from) should be write in this paper

Line 194: should be have the equation

Line 198: GC (what series of GC, what country)  

Line 202:  should be rewrite and make it easy to understand

“The solution was hated at 80 C for 2 hours……………”

Line 204: Is the organic phase same with the supernatant or not?

Line 205: add well after mixed >>> mixed well

Line 211: at 250 C, at 200 C >>> add at

Line 212: for 5 min not minutes, because you use min before and keep to use the same pattern

Line 219: (2 % vol/vol) no need comma

Line 237: A 5 g of sample……………………….

Line 261: (0.15 x kg milk x % Fat)

Line 265: Could you please rewrite the sentence >>> The calculation of health lipid indices were used by the equation of Ulbricht and Southgate [30].

Line 296: Table 1. Ingredients …………………………………………………….. of experimental diets

Line 296: see in the table Crude protein should be write CP and Crude fat should be write CF and explain the abbreviation under the table ** if you use abbreviation have to use the same pattern or you can write the full

The last part of Table 1. Add of  >>>>> Fat-soluble antioxidants (mg/kg of DM)

Table 2. recheck to add “s” the same with Table 1.

Table 3. Should be check abbreviation of crude protein and crude fat with the same pattern as comment before

Line 484: approximately 6% of DM,………………………….

Line 292-293: please rewrite the sentence to formal for the research paper it will be nice

This research is interesting and you have to modified the introduction and all comments and have to add some reference in update too , almost your reference is not update so need to add more update reference. 

Have a good luck for all authors

Author Response

Dear reviewer,

Thank you very much for your time and the all the valuable comments on our manuscript. We have dealt with all the concerns raised and revised the manuscript accordingly. All suggested changes have been highlighted in the full-text. Thus, a revised version of the manuscript is submitted.

Thank you again for your work.

Regards,

Mierlita Daniel

Reviewer 2 Report

The aim is clear and interesting, though I suggest to reformulate it. Similar researches were carried out with Holstein cows and those studies should be cited in this manuscript.
The sentence L118-120 should be removed/reformulated taking into account my suggested literatures. At least these should be included in the introduction as well as in the discussion part of the manuscript.

The set up of the experiment is clear, the used statistical methods are relevant.

The usage of Greek letters is confusing. E.g.: There are Greek letters in table 2. Alpha (L317) and gamma (L320) were written whereas the Greek letter was used for alpha (L323)  and beta (L322). This must be uniform in the manuscript.

The literatures below should be taken into account during revising the manuscript:
- https://doi.org/10.1017/S175173110900442X
- http://dx.doi.org/10.15666/aeer/1602_15531562

Author Response

Dear reviewer,

Thank you very much for your time and all the valuable comments on our manuscript. We have dealt with all the concerns raised and revised the manuscript accordingly. All suggested changes have been highlighted in the full-text. Thus, a revised version of the manuscript is submitted.

Thank you again for your work.

Regards,

Mierlita Daniel

Round 2

Reviewer 1 Report

This paper is interesting, and value parameter but have to follow all coment also recheck through the menuscritp, the comment attached below.

Comments [Round 2]

Topic: The Effects of Feeding Milled Rapeseed Seeds with Different 3 Forage: Concentrate Ratios in Jersey Dairy Cows on Milk Production, Milk Fatty Acid Composition, and Milk Antioxidant 5 Capacity.

Line 25: add “s” after profile from your study fatty acid improved more than one fatty acid.

Line 39: It is complicated, should be delete cow

Line 40: status or capacity? Which one better and formal

Line 232: should be change (25, 26 and 27 d)

Line 731: g/day which one you gonna use, day or d

Line 754: often use our research, can change to this study, this research , this experiment

Please recheck “s” Fatty acids profile or fatty acid profiles

Should be use the same word

Author Response

(The authors gave the same response as above.)
